# Recent Advances of Ambient Mass Spectrometry Imaging and Its Applications in Lipid and Metabolite Analysis

**DOI:** 10.3390/metabo11110780

**Published:** 2021-11-15

**Authors:** Keke Qi, Liutian Wu, Chengyuan Liu, Yang Pan

**Affiliations:** National Synchrotron Radiation Laboratory, University of Science and Technology of China, Hefei 230029, China; kkq77@ustc.edu.cn (K.Q.); liutianw@mail.ustc.edu.cn (L.W.)

**Keywords:** ambient mass spectrometry imaging (AMSI), lipid, metabolite

## Abstract

Ambient mass spectrometry imaging (AMSI) has attracted much attention in recent years. As a kind of unlabeled molecular imaging technique, AMSI can enable in situ visualization of a large number of compounds in biological tissue sections in ambient conditions. In this review, the developments of various AMSI techniques are discussed according to one-step and two-step ionization strategies. In addition, recent applications of AMSI for lipid and metabolite analysis (from 2016 to 2021) in disease diagnosis, animal model research, plant science, drug metabolism and toxicology research, etc., are summarized. Finally, further perspectives of AMSI in spatial resolution, sensitivity, quantitative ability, convenience and software development are proposed.

## 1. Introduction

Mass spectrometry imaging (MSI) is a powerful analytical method, which is able to visualize the spatial distribution of a large number of compounds from the complex sample surface in a single experiment [1].

Generally in MSI experiments, the sample sections should be carefully prepared and then scanned and ionized by various desorption/ionization methods. The ion intensity of each individual compound at the target mass-to-charge ratio (*m*/*z*) are extracted from each pixel’s mass spectrum and combined into a heat map revealing the relative distribution of that compound throughout the sample surface. Compared with conventional tag-probe labeling optical imaging methods, MSI enables the un-targeted imaging of multiple compounds without the need for labeling.

Ambient mass spectrometry refers to those ionization techniques operated in an atmospheric environment with little or no sample preparation [2,3]. It was firstly introduced by Cooks et al. in 2004 with the invention of desorption electrospray ionization (DESI) [4]. Due to its high sensitivity, high speed and easy operation at native conditions, ambient mass spectrometry was widely used in MSI, and ambient mass spectrometry imaging (AMSI) has been developed to be an important branch of MSI. In AMSI, compounds are desorbed from the sample surface at ambient conditions, ionized by charged microdroplets, photons or plasma, and then introduced into the mass spectrometer for further detection. Up to now, AMSI techniques based on different ionization methods have been proposed for the improvement of sensitivity and spatial resolution, and they have been widely applied in disease diagnosis, drug metabolism, toxicology research, forensic investigation and plant science [5,6,7]. Lipid and metabolite are the small-molecule entities that have key roles for the establishment of physiological function within the biological systems. The MSI of a global lipid and the metabolite profile from a biological tissue can help with an enhanced understanding of disease molecular mechanisms, the discovery of biomarkers and the elucidation the mechanisms of drug action [8].

Several excellent reviews on different topics of AMSI have been reported. For example, Xue et al. summarized AMSI techniques from the aspects of ion source devices, ionization mechanism, resolution, sensitivity and applications in 2019 [9]. Xiao et al. introduced the important applications of the AMSI technique in pharmacology, drug metabolism, clinical diagnosis and toxicological evaluation in 2020 [10]. In this review, we will summarize the developments of AMSI technologies according to one-step/two-step ionization strategies and their application advances in lipid and metabolite from 2016 to 2021. In addition, the prospects of AMSI techniques and their applications for biological samples in the near future are discussed. Figure 1 shows the schemes of AMSI for lipid and metabolite analysis in this work.

## 2. Development of AMSI Techniques

In AMSI, the process of target analytes on the sample surface being desorbed and simultaneously ionized is called the one-step ionization strategy, whereas when the desorbed analytes are post-ionized by another ionization source this is called the two-step ionization strategy.

### 2.1. One-Step Ionization Strategy

#### 2.1.1. Desorption Electrospray Ionization (DESI) 

DESI was the first ambient desorption ionization technique operated in ambient conditions [4], and was used in MSI from 2006 [11]. DESI is a representative one-step ionization source, where the analyte molecules on the surface are desorbed and ionized within one single step by high-speed, charged microdroplets generated by electrospray ionization (ESI) (Figure 2A) [12].

DESI has great advantages for the analysis of lipids, metabolites and other compounds in biological tissues at ambient conditions without any sample pretreatment. However, improving its sensitivity is still challenging [13,14]. Recently, Abliz’s group developed a new AMSI technique termed airflow-assisted desorption electrospray ionization (AFADESI), which is based on DESI (Figure 2B). This method has a picomolar sensitivity for targeted drug molecules by virtue of high transport efficiency with the assistance of airflow, extended length and area of sampling, and it was successfully applied to the imaging of large, whole-body sections [15,16,17]. AFADESI could improve the signal of lipids and drugs by 2~25-fold by washing the whole body tissue sections of mice with in situ hydrogel [18]. Varying the spray solvent is a convenient method to improve sensitivity. Wang et al. added trifluoroacetic acid into DESI spray solvent to increase the ionization efficiency of cholesterol and other metabolites by 21~62-fold [19]. Another shortfall of DESI is its relatively low spatial resolution (around 50–200 μm). By optimizing experimental parameters such as capillary size, the composition and flow rate of spray solvent, mass spectrometry scan rate and step size, a resolution of 35 μm in the analysis of mouse brain could be obtained [20]. In addition, a new ionization method working in a similar way to DESI, named easy ambient sonic spray ionization (EASI), without high voltages, electrical discharges, UV and laser beams was also used for AMSI (Figure 2C) [21,22,23].

#### 2.1.2. Desorption Atmospheric Pressure Chemical Ionization (DAPCI) and Low-Temperature Plasma (LTP)

In addition to charged microdroplets, plasma is another good choice to desorb and ionize the analytes from the sample surface in open air. DAPCI was introduced as a plasma-based ambient ion source by Cooks’ group in 2005 (Figure 2D) [24]. Ouyang et al. distinguished the human squamous cell carcinoma lung cancer from normal tissues using phosphatidylcholine (PC) and sphingomyelin (SM) profiles using DAPCI-MSI [25]. LTP is another plasma-based technique (Figure 2E) [26]. By virtue of the low ion source temperature (<40 °C), LTP enables non-invasive MSI without thermal damage to the sample. Liu et al. applied the LTP-AMSI to visualize the distribution of seal patterns on calligraphy and paintings and to distinguish genuine seals from counterfeit examples with a spatial resolution of 250 µm [27].

**Figure 2 metabolites-11-00780-f002:**
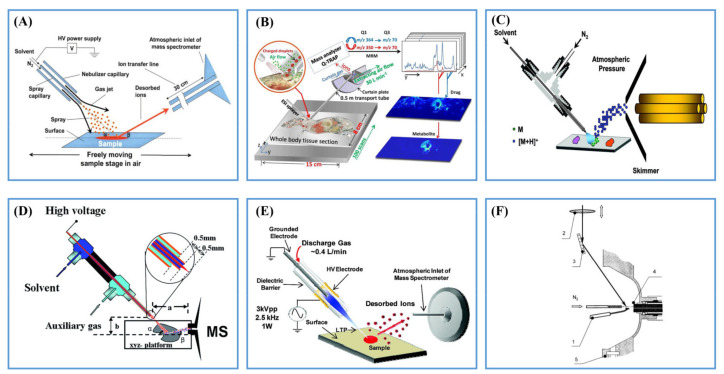
(**A**) Scheme of DESI ion source. Reprinted with permission from the authors of [4]. Copyright (2004) American Association for the Advancement of Science. (**B**) Scheme of AFADESI ion source. Reprinted with permission from the authors of [16]. Copyright (2013) American Chemical Society. (**C**) Scheme of EASI ion source. Reprinted with permission from the authors of [21]. Copyright (2008) American Chemical Society. (**D**) Scheme of DAPCI ion source. Reprinted with permission from the authors of [25]. Copyright (2017) Royal Society of Chemistry. (**E**) Scheme of LTP ion source. Reprinted with permission from the authors of [26]. Copyright (2008) American Chemical Society. (**F**) Scheme of AP-MALDI ion source. Reprinted with permission from the authors of [28]. Copyright (2000) American Chemical Society.

#### 2.1.3. Atmospheric Pressure Matrix-Assisted Laser Desorption/Ionization (AP-MALDI)

Matrix-Assisted Laser Desorption/Ionization (MALDI) is the most widely used MSI method, where laser generated photons desorb and ionize analytes on the sample surface with the assistance of a matrix under vacuum conditions. Atmospheric pressure MALDI, i.e., AP-MALDI, operated at ambient pressure, was first introduced by Laiko et al. in 1998 [28]. Compared with vacuum MALDI, AP-MALDI can be coupled with many MS instruments with little modification [29]. Another attractive feature of AP-MALDI is that the small laser spot size allows it to have higher resolution and better repeatability than desorb/ionization methods based on charged microdroplets and plasma. A high lateral resolution of 1.4 µm at native physiological conditions was obtained by improving the laser focus diameter in reflective geometry (Figure 2F) [30]. Table 1 summarizes the characteristic of six commonly used one-step desorption/ionization techniques for AMSI.

### 2.2. Two-Step Ionization Strategy

In the two-step ionization strategy, desorption and ionization of analytes are separated into two steps: (1) generating analyte-containing droplets/particles/gastification products from target samples; (2) post-ionizing the desorbed neutral species. The first step is normally fulfilled by thermal desorption, laser desorption and droplet pick-up, etc. As is well known to us, some compounds can be ionized during the initial laser desorption and microdroplets pick-up processes. However, due to the matrix effect in the microenvironment of biological tissues, most of the desorbed molecules are not ionized [31]. In the second step, the desorbed neutral species can be post-ionized by using charged microdroplets, plasma or photons in ambient conditions.

#### 2.2.1. Post-Ionization by ESI

Figure 3 shows four typical two-step ionization strategies using ESI as the post-ionization method. In 2010, Laskin et al. introduced nanospray desorption electrospray ionization (nano-DESI), where analytes on the sample surface were extracted using a liquid bridge created by a primary capillary and then ionized by nanospray ionization in the second capillary emitter (Figure 3A) [32]. Yin et al. successfully used the nano-DESI to visualize the distribution of a variety of lipid classes in pancreatic islets of mice [33]. The spatial resolution of nano-DESI MSI could be improved three-fold by over sampling [34]. In order to image biological samples of complex topography, a shear-force probe was integrated with nano-DESI for constant-distance mode AMSI [35,36] and the visualization of chemicals, and the topographic of living bacillus subtilis ATCC 49760 was demonstrated with a high spatial resolution of 12 μm [35]. The liquid bridge for sampling extraction could also be formed by a single probe, which was developed by Yang’s group in 2015 (Figure 3B). Single probe ionization has been used in the MSI of tissue sections such as the brains and kidneys of mice, with minimal sample preparation, at 8.5 μm high spatial resolution [37,38]. Moreover, the single probe ionization broadens the application scope of ambient mass spectrometry technique in single-cell analysis under ambient conditions [39,40].

Due to its high power density, high stability and high brightness, lasers are a promising tool for sample ablation and desorption. Compared with liquid bridge extraction, laser ablation enables sampling in a confined area with high repeatability and throughput. Laser ablation electrospray ionization (LAESI) (Figure 3C) was introduced by Vertes’s group in 2007 [41]. The analytes were ablated and desorbed from the sample surface by a mid-IR laser (2.94 μm) to form a plume of gaseous molecules, which moved upward and made contact with an orthogonal electrospray plume to produce the ions. As the O-H bond in the water molecule can strongly absorb mid-IR light [42], LAESI is especially suitable for analyzing samples containing water molecules under ambient conditions [43]. Therefore, Kulkarni et al. applied the LAESI-MSI to map and compare the root metabolome of different plant tissues [44]. LAESI MSI was also used for the 2D and 3D molecular imaging of microbial culture and the study of the inhibition of antibiotica in 2016 [45]. Matrix-assisted laser desorption electrospray ionization (MALDESI) was first introduced in 2006 by Muddiman et al. [46]. The mechanism of desorption/ionization is similar to LAESI; a 337 nm UV and 2.94 μm IR laser (IR-MALDESI) are ultilized to vaporize molecules on the sample surface, which are then post-ionized using ESI [47]. MALDESI and IR-MALDESI use glycerol, 2,5-dihydroxybenzoic acid (2,5-DHB), sinapinic acid, glycerol, succinic acid, endogenous water and exogenous ice as the matrix to form the co-crystallized spots of matrix analyte for better desorption of compounds from the sample surface in ambient conditions [47,48]. Compared to tissues relying solely on their endogenous water content, the ion abundances were improved by nearly one order of magnitude by using a layer of deposited ice as an energy-absorbing matrix in IR-MALDESI MSI. In order to prevent the condensation of water on the tissue, which could potentially delocalize lipids and metabolites on the tissue, the tissue was maintained at −9 °C while under nitrogen purge and then an exogenous ice layer was rapidly deposited on the sample plate at ambient relative humidity (>10%) [47,49]. Compared to an organic matrix, water and ice can reduce the interference caused by an organic matrix at a low-mass range, which has great advantages in the analysis of small compounds, especially lipids and metabolites. The MALDESI and IR-MALDESI-MSI techniques have been applied to map the distribution of compounds in the mouse bones [50], cherry tomatoes [51], cancerous hen ovarian tissues [52], skin tissues [53], over-the-counter pharmaceuticals [54], 3D imaging [54] and lipid and metabolites analysis [55]. Recently, Lee et al. developed a new ambient ion source named laser desorption/ionization droplet delivery (LDIDD) with a high resolution (2.4 to 3 μm) by using 266 nm UV laser beams (Figure 3D) [56]. The ablated analytes were simultaneously captured, delivered and ionized by a spray of liquid microdroplets. The single cells of apoptotic HEK cells were analyzed and a difference of fatty acids and lipids between HEK cells undergoing apoptosis was observed.

**Figure 3 metabolites-11-00780-f003:**
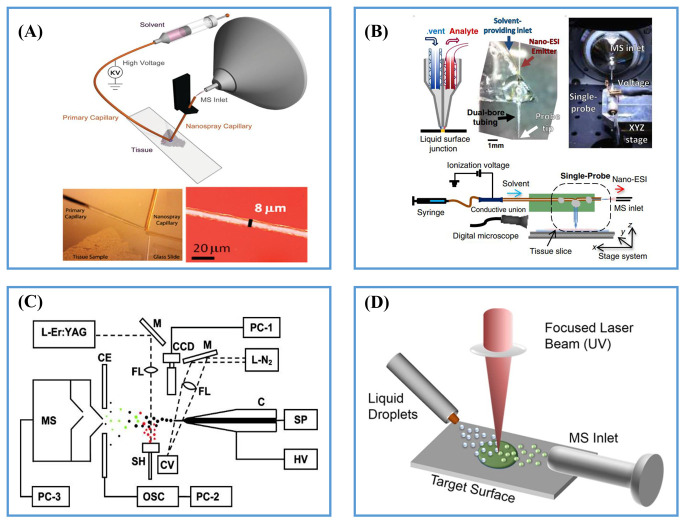
(**A**) Scheme of nano-DESI ion source. Reprinted with permission from the authors of [57]. Copyright (2012) American Chemical Society. (**B**) Scheme of single probe ion source. Reprinted with permission from the authors of [38]. Copyright (2015) American Society for Mass Spectrometry. (**C**) Scheme of LAESI ion source. Reprinted with permission from the authors of [41]. Copyright (2007) American Chemical Society. (**D**) Scheme of LDIDD ion source. Reprinted with permission from the authors of [56]. Copyright (2016) American Chemical Society.

#### 2.2.2. Post-Ionization by Plasma Ionization

Plasma is also widely used as an ionization source in ambient mass spectrometry [58]. In 2012, Liu et al. introduced plasma-assisted laser ionization (PALDI) [59], which combines the ion sources of laser sampling and direct analysis in real time (DART) (Figure 4A). A pulsed Nd: YAG (355, 532 and 1064 nm, 10 Hz, and a pulse duration of 10 ns) was set at 45° incident angle to form a plume of neutral analytes, which were then ionized by the heated metastable plasma of excited ions generated by DART through the reactions of proton transfer, charge exchange and penning ionization. Feng et al. used PALDI to analyze the active compounds in traditional Chinese medicines (Radix Scutellaria) with a resolution of 60 μm [60]. Recently, Zenobi et al. constructed a novel AMSI platform named laser ablation dielectric barrier discharge ionization (LADBDI) with a low-cost, flexible strategy and high lateral resolution (20 μm) (Figure 4B) [61]. Recently, they introduced a fiber probe laser ablation (FPLA) to dielectric barrier discharge ionization termed FPLA-DBDI and a higher spatial resolution of ~5 μm was obtained [62].

In 2017, Musah et al. developed another plasma post-ionization-based AMSI termed laser ablation direct analysis in real time (LADI) [63], which combined a 213 nm Nd: YAG solid state UV laser and DART (Figure 4C). This platform enabled facile determination of the spatial distribution of small molecules spanning a range of polarities in a diversity of sample types and required no matrix, vacuum, solvent or complicated sample pretreatment steps. Hardering et al. introduced laser ablation atmospheric pressure chemical ionization (LAAPCI) in 2014 [64], where analyte molecules were irradiated and desorbed by the ns pulsed Nd: YAG laser (213 nm) to form a fine aerosol and then transported into an APCI mass spectrometer for post-ionization by a corona discharge (Figure 4D). In 2020, plasma was used as an in-line, post-ionization method to improve the ionization efficiency of certain compounds in AP-MALDI by Elia et al. [65].

#### 2.2.3. Post-Ionization by Photoionization (PI)

Owing to the favorable characteristics of soft ionization, no polarity discrimination and reduced ion suppression compared with ESI, PI is attracting increasing attention in ambient mass spectrometry. In 2007, desorption atmospheric pressure photoionization (DAPPI) was introduced by Kauppila et al. (Figure 5A) [66], where a heated jet of vaporized solvent containing dopant (e.g., toluene) from a heated nebulizer microchip was directed toward the sample surface to desorb the analytes, and photons emitted from a vacuum ultraviolet (VUV) discharge lamp were used to ionize desorbed analytes [67]. As analytes were desorbed by the heated solvent vapor, the thermally unstable compounds were difficult to analyze and the thermal desorption area (1000 μm) was very large, which seriously restricted the application of DAPPI in AMSI [68]. In 2012, Vertes et al. introduced a novel atmospheric pressure ion source for in situ AMSI, named laser ablation atmospheric pressure photoionization (LAAPPI), where an infrared (IR) laser running at a wavelength of 2.94 μm was used to ablate the analyte molecules from the sample surface (Figure 5B). The ablated analytes were ionized by a VUV krypton discharge lamp (118 nm) [69]. A 400 μm lateral resolution could be obtained using LAAPPI-MSI for the imaging of triterpenoids at *B. pendula* lenticels [70]. In 2017, Hieta et al. improved the spatial resolution by increasing the distance between the laser and the focusing lens, and a spatial resolution of 60 μm could be obtained for the MSI of mouse brains by LAAPPI [71]. In 2020, this IR laser-beam-focusing technique was applied for sub-100 μm spatial resolution in LAAPPI and LAESI-MSI with high sensitivity, good spot-to-spot repeatability and high robustness by optimizing operational parameters [72]. Photoionization was also used for post-ionization for enhanced sensitivity. In 2015, Soltwisch et al. used a wavelength-tunable, post-photoionization laser to initiate secondary MALDI-like ionization processes in the gas phase and the ion yields for numerous lipid classes, saccharides and liposoluble vitamins that increased by up to two orders of magnitude within the 5 μm wide laser spot [31]. In ambient DESI, a variety of species, especially nonpolar compounds in biological tissue, were hard to ionize due to the polarity discrimination of ESI-like mechanisms and a strong ion suppression effect caused by salt and polar lipids. In 2019, Liu et al. proposed a compact post-photoionization assembly to combine with DESI (DESI/PI) for the simultaneous imaging of polar and nonpolar compounds in biological tissues (Figure 5C) [73]. The compounds on the tissue surface were desorbed and ionized by a DESI sprayer, introduced into a heated transfer tube and post-ionized by a coaxially oriented krypton DC discharge VUV lamp with photon energies of 10.0 and 10.6 eV. Signal intensities of more than two orders of magnitude higher for certain biomolecules, such as cholesterol, creatine, galactosylceramide (GalCer) lipids and neutral catechins, were obtained by DESI/PI. More importantly, DESI/PI is a portable and compact ambient ionization source, which can be connected to other MS inlets with minor modifications. Table 2 summarizes the properties of twelve commonly used two-step ionization techniques for AMSI.

## 3. Applications in Lipids and Metabolites

### 3.1. Lipids

Lipids are components of the cell membrane, and they play a vital role in cell membrane fluidity, neurotransmitter transmission and transport and energy supply [74,75]. Lipid compositions can reflect histological type and cell growth state; hence, the alteration of lipid metabolism is linked to the occurrence of several human diseases [76], such as Alzheimer’s disease, breast cancer [77] and basal cell carcinoma [14]. Lipidomic analysis can provide valuable information for understanding the molecular pathological mechanisms of many diseases, diagnosis and differentiation of diseases and assessment of resection margins during clinical surgery, etc. [6,78,79,80]. It should be noted that the spatial distribution of proteins could also be visualized by the fluorescent labeling method, termed immunohistochemistry and immunofluorescence, whereas few other technologies can image lipids [81]. In past years, it has been demonstrated that AMSI techniques can be performed to visualize the spatial distribution of the sample surface compounds in the native state, and that they have a high sensitivity to lipids and other small molecules in diseased tissues, animal models and plants, etc.

Cancer cells will reprogram their metabolic pathways to meet the abnormal energy requirements of proliferation. Molecular heterogeneity can lead to a variety of cancer-related clinical behaviors. Compared with conventional histopathology, MSI can provide abnormal distribution of lipid metabolism in diseased tissues. Spatial distribution of lipids in different organs and tissues has important clinical significance in the diagnosis of diseases and can provide information about tumor margins to guide doctors to remove corresponding tumor areas during surgery [6]. In addition, understanding the molecular spatial information of cancer tissues will help us to further study the molecular pathological mechanism of cancer and find possible treatment methods [82]. AMSI has been used to investigate the lipid profiles of human serous ovarian tumors [83], breast cancer [84], brain tumors [85], lung cancer [86], prostate cancer [87], human basal cell carcinomas [14], diabetic kidney disease [88], epithelial ovarian carcinomas [89], cerebral ischaemia [90], epithelial ovarian carcinomas [89], pancreatic cancer [80], clear renal cell carcinomas [91] and muscle invasive bladder cancer [92]. In 2016, the Eberlin group utilized DESI AMSI to investigate the lipid profile of mitochondria-rich thyroid oncocytic tumors and the distribution of cardiolipins (CLs) overlapped with regions of mitochondria-rich oncocytic cells [93]. Their experiments showed that CLs were the new biomarkers for clinical and therapeutic targetsof thyroid oncocytic tumors. Recently, DESI was used for the MSI of 103 samples of breast cancer tissue from the United States and Brazil. The least absolute shrinkage and selection operator (Lasso) classification model and principle component analysis (PCA) demonstrated that DESI-MSI was a robust and reproducible technique for the diagnosis of breast cancer tissue using lipid profiles (Figure 6A) [94]. In 2021, Qi et al. applied the DESI and DESI/PI-MSI to visualize the distribution of small molecules of human melanocytic nevi, and more than 108 polar and nonpolar lipids were found to be specifically distributed (Figure 6B) [95]. Moreover, cholesterol was identified to be a potential biomarker of melanocytic nevi using PCA analysis and the immunohistochemistry (IHC) experiment.

Precise diagnosis and differentiation of different cancers along with subtyping are significant in avoiding overtreatment during surgery. AMSI has been utilized for visualizing the lipid signatures of human cancer subtypes. In addition, the in situ visualization of various lipids in biological tissue by AMSI could help with the investigation of pathological mechanisms and the discovery of biomarkers of various diseases. It is important to distinguish invasive breast ductal carcinomas (IDC) and ductal carcinomas in situ (DCIS) in the process of surgery. In 2016, Mao et al. successfully discriminated against various subtypes and grades of DCIS and IDC by analyzing lipids using AFAI-MSI. They demonstrated that phospholipids were more abundant (PCs) in IDC tissue, whereas fatty acids (FAs) were more accumulated in DCIS tissue (Figure 6C) [96]. Santoro et al. used in situ DESI-MSI to examine the lipid profiles of breast cancer molecular subtypes and precursor lesions in 2020 [77]. Their findings revealed that polyunsaturated fatty acids, deprotonated glycerophospholipids and sphingolipids were the top ions in invasive breast cancer (IBC), highly saturated lipids and antioxidant molecules could have the ability to distinguish IBC from adjacent benign tissue (ABT). IBC and ductal carcinomas in situ (DCIS) were distinguished by cell signaling and apoptosis-related ions, including fatty acids and glycerophospholipids. These findings provided a new insight in understanding the pathogenesis of breast cancer. Zhang et al. used Lasso to successfully discriminate various types of renal tissue, including normal kidney, renal oncocytoma and three subtypes of renal cell carcinomas (RCC) from 71 patient samples [97]. In 2020, Bensussan et al. distinguished the non-small cell lung cancer subtypes by virtue of lipid profile using DESI-MSI [98]. Besides the study of lipid profiling in different subtypes of cancer, AMSI techniques also showed a great potential in visualizing the distribution of lipids in malignant samples from benign lesions. In 2018, Margulis et al. analyzed 86 human basal cell carcinoma (BCC) specimens collected in Mohs surgery to find the alterated numerous metabolites and lipids in BCC and normal human skin. Compared to normal tissues, BCC nets were distinctively abundant in fatty acids (arachidonic acid, oleic acid and palmitic acid, etc.), glycerophosphoglycerol (PG 34:1), glycerophosphoserine (PS 36:1) and glycerophosphoinositol (PI 38:4) (Figure 6D) [14]. The level of fumarate, a key intermediate of the Krebs cycle, was markedly depleted in the BCC tissue and played an important role in the classification of BCC and normal tissue for Lasso analysis.

AMSI was also engaged in studying molecular signatures in animal models. In 2018, the Zare group combined DESI-MSI with machine learning to recognize the molecular feature of myocardial infarction from in vivo mouse models and 62 molecular ions were selected for the classification of cardiac tissue with a high average accuracy, recall and precision, i.e., 97.4%, 95.8% and 96.8% [99]. In 2019, the Lanekoff group found a significant increase in several 18-carbon unsaturated non-esterified fatty acids, monoacylglycerols and short- and long-chain acylcarnitines in diabetic kidney tissue by ambient nano-DESI MSI [100]. In 2021, Zeng et al. combined systematic metabolomics analysis and AP-MALDI MSI and revealed the lipid biomarkers in female ICR mice under cadmium exposure (Figure 6E) [101]. Significant lipid alterations in the liver, heart and kidney were observed and a new insight for deeply understanding the toxicological mechanism of cadmium was provided. Zebrafish is another widely used animal model in biology, toxicology, pharmacology, medicine and limb regeneration. In 2018, Kim et al. studied the molecular spatial distribution changes that occurred in the regeneration of fresh zebrafish caudal fins by using newly developed atmospheric pressure-nanoparticle and plasma-assisted laser desorption ionization (AP-nanoPALDI) (Figure 6F) [102]. Compared to the original area in the early stages, the small molecules including lipids, amino acids and neurotransmitters were more evenly distributed throughout the bony rays and inter-ray mesenchymal tissues. In 2018, Liu et al. applied AP-MALDI to image whole zebrafish sections exposed to fipronil and significant chemical fingerprint differences in phospholipids between exposed and untreated zebrafish groups were observed [103].

Moreover, the AMSI technique has been used as an active tool for the imaging of lipid isomers with different double-bond positions. In 2018, Klein et al. integrated DESI and ultraviolet photodissociation to characterize the lipid double-bond location isomers in different tissue sections [104]. The relative abundance of lipid isomers in specific tissues, including endometrial tissue, kidney tissue, brain tissue and human lymph node tissue, containing thyroid cancer metastasis were mapped. For the rapid MSI of unsaturated lipids at an isomeric level, Zhang et al. proposed a simple method to accelerate the oxidation of C=C isomers of unsaturated lipids in air by using moist heating (Figure 6G) [105]. After only 10 min of accelerated oxidation, the unsaturated lipid isomers in nude lung cancer tissue and human thyroid cancer tissues were successfully imaged by AFADESI. In 2021, for lipid isomer identification, Unsihuay et al. reported an online photochemical derivatization approach, i.e., ^1^O_2_ reacted with lipids to form lipid hydroperoxides (LOOHs), which was then followed by collision-induced dissociation (CID) to obtain diagnostic peaks (Figure 6H) [106]. The Paternò–Büchi (PB) reaction is another commonly used method for identifying the location of C=C double bonds in unsaturated lipids and simplified PB reaction conditions were introduced by spraying the reagent solution on the tissue surface with only 10 min preprocessing [107].

**Figure 6 metabolites-11-00780-f006:**
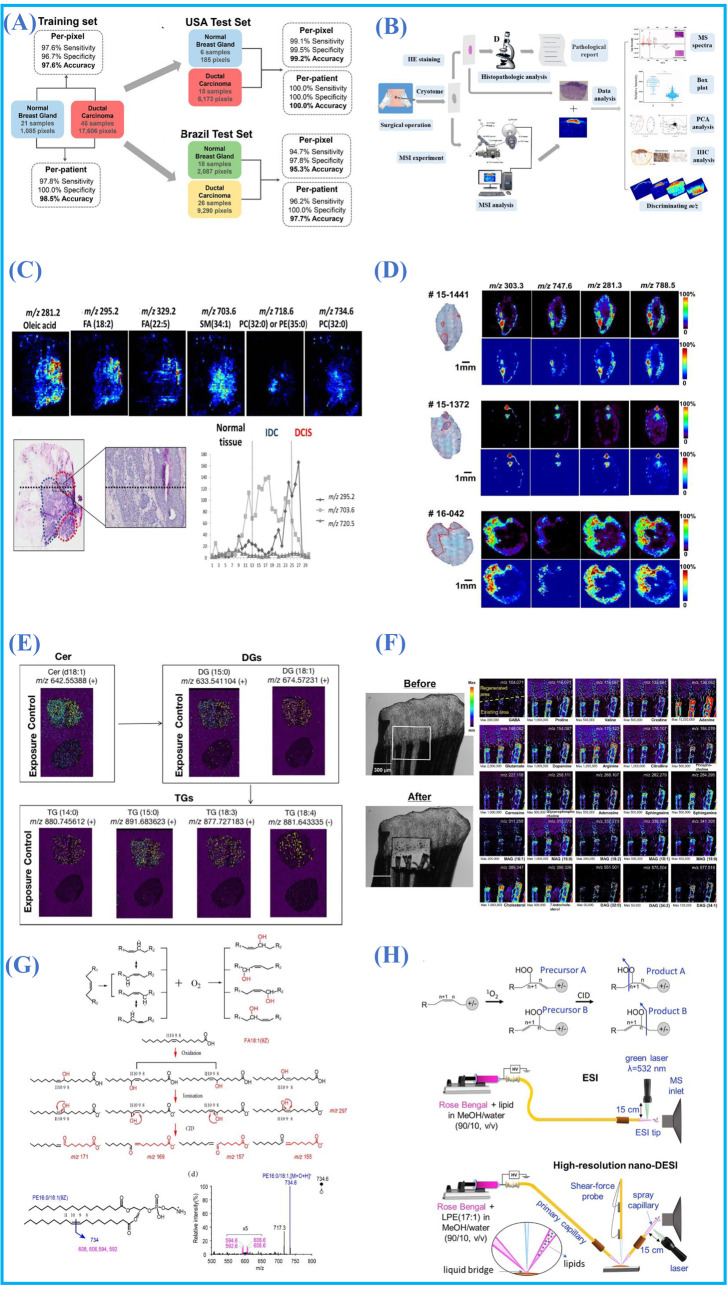
(**A**) Classification of per-pixel and per-patient prediction results from normal and IDC samples. Reprinted with permission from the authors of [94]. Copyright (2018) American Chemical Society. (**B**) The proposed workflow for diagnosis of melanocytic nevi. Reprinted with permission from the authors of [95]. Copyright (2021) Elsevier B.V. (**C**) Representative compounds MSI images and H&E staining of normal tissue, IDC and DCIS samples. Reprinted with permission from the authors of [96]. Copyright (2016) Springer Nature. (**D**) Selected 2D MSI ion images and H&E stained BCC and normal tissue in three specimens. Reprinted from [14] with the permission of (2018) National Academy of Sciences. (**E**) AP-MALDI MSI images of representative lipids from liver tissues of mice under cadmium exposure. Reprinted with permission from the authors of [101]. Copyright (2021) Elsevier B.V. (**F**) Representative MSI ion images and optical images of zebrafish caudal fin. Reprinted with permission from the authors of [102]. Copyright (2018) American Chemical Society. (**G**) Mechanism of the oxidation of unsaturated lipids. Reprinted with permission from the authors of [105]. Copyright (2021) American Chemical Society. (**H**) Mechanism experiment setup of the online single oxygen reaction with lipids. Reprinted with permission from the authors of [106]. Copyright (2021) Wiley-VCH GmbH.

### 3.2. Metabolites

Molecular metabolites such as neurotransmitters, amino acids and vitamins play an important role in biosynthesis, energy production and supply, signal transduction and regulation and cognitive processes [108,109,110]. Changes in small molecules metabolites are often closely related to the nervous system and disease states, such as depression, Alzheimer’s disease, movement disorders, being overweight, obesity and so on [111,112,113,114,115]. Therefore, a comprehensive and detailed understanding of the relative abundance and spatial distribution of small-molecule metabolites in organisms is an outstanding contribution to further understanding the metabolic reorganization of tumors, elucidating the metabolic mechanism in the process of disease development and searching for potential metabolic markers for disease diagnosis. As an unlabeled molecular imaging method, AMSI technique can obtain spatial distribution information of many small-molecule metabolites in a single experiment with little or without any pretreatment. For the imaging of labile metabolites in ambient conditions, the labile group could be protected via in situ chemical derivatization [116].

In 2017, Banerjee et al. found a marked metabolic derangement especially in Krebs cycle intermediates in malignant prostate cells compared to benign counterparts using DESI-MSI [87]. In 2019, Zeper et al. used AFADESI-MSI for in situ metabolomics analysis to localize the tumor-associated metabolites in their native state from 256 esophageal cancer patients. Metabolic pathways of various metabolites, such as proline biosynthesis, glutamine metabolism, uridine metabolism, histidine metabolism, fatty acid biosynthesis and polyamine biosynthesis, were mapped. Moreover, six abnormally expressed enzymes including pyroline 5-carboxylate reductase 2 (PYCR2) and uridine phosphorylase 1 (Upase 1) that are associated with metabolic pathways were found to be altered in esophageal squamous cell carcinomas (Figure 7A) [117]. In 2021, more than hundreds of different polar metabolites involved in multiple metabolic pathways, such as neurotransmitters, purines, polyamines, cholines, organic acids and carbohydrates, were mapped in rat brains by AFADESI-MSI. Moreover, this method was successfully used to discover the altered metabolites of the scopolamine-treated Alzheimer’s model [118]. In 2020, Niziol et al. introduced a new AMSI technique termed infrared laser ablation-remote-electrospray ionization (LARESI) to localize the metabolites in human kidney tissues [119]. The abundance of ten amino acids, four nucleotides, nucleobases, lactate and vitamin E varies in renal cell carcinomas, and normal tissue was obtained with the help of LARESI-MSI and tandem mass spectrometry. In 2021, Song et al. reported a new MSI technique called hydrogen–deuterium exchange DESI MSI (HDX-DESI-MSI) to detect the pH value and visualize the acidic tumor microenvironment (TME) (Figure 7B) [120]. In this work, methanol-D_2_O was used as the spray solvent of DESI and a xenograft tumor was roughly divided into three microregions with different pH values, including 6.4 ± 0.2, 6.8 ± 0.2 and 7.2 ± 0.2. Further multivariate statistics and pathway enrichment analysis showed that sulfonic acid and fatty acids contribute to regional acidity, aminoacyl-tRNA, unsaturated fatty acids, arginine and aromatic amino acids were found in the acidic stroma region. Yan et al. successfully visualized 107 metabolites in astrocytes and neurons by combining DESI-MSI and immunofluorescence staining [121].

Visualizing the distribution of metabolites in plants is also of great significance for further understanding their functions, biosynthesis pathways, possible transport and metabolic mechanisms [7]. In recent years, with the progress in the ambient mass spectrometry technique, it is possible to use the AMSI technique to image small molecules and various metabolites in plant tissues with high specificity. The research of AMSI techniques in the plant field mainly focuses on the distribution of active compounds and metabolites in medicinal plants and food crops. What is more, the changes in spatial distribution of compounds in plants with different growing status, i.e., immature, wound and disease are also the research hotspot. Compounds containing the information about spatial distribution are commonly captured by a versatile substrate for reliable MSI in plant tissues, which differ from animal tissues. Many imprinted materials were used to investigate the distribution of a diverse variety of metabolites and species, such as thin layer chromatography plates, polytetrafluoroethylene, porous silicon, etc. In 2016, Hemalatha et al. introduced a new material, termed electrospum nylon-6 nanofiber mats, for a smart surface for imprint imaging in DESI-MSI (Figure 7C) [122]. Due to its large surface area, the electrospum nanofiber can quickly capture the diverse classes of metabolites during imprint imaging under ambient conditions. In 2017, Enomoto et al. visualized the distribution of abscisic acid (ABA) and 12-oxo-phytodienoic acid (OPDA) in immature phaseolus vulgaris L.seeds using DESI (Figure 7D) [123]. In 2021, Zhang et al. investigated the phytohormones from wounded arabidopsis leaves imprinted on a thin-layer chromatography plate using DESI-MSI and the LC/MS experiment [124]. Higher levels of salicylic acid, jasmonates, abscisic acid and indole-3-acetic acid were found in wounded leaves compared to control leaves, which was consistent with the result of LC-MS. In 2020, de Moraes Pontes et al. used DESI-MSI for the diagnosis of huanglongbing disease and metabolites including abieta-1,11,13-trien-18-oic acid and 4-acetyl-1-methylcyclohexene, which mainly accumulate in symptomatic leaves [125]. AMSI was also used to investigate fungus-related diseases, and several phospholipids and other unidentified metabolites in fungus and oomycete pathogens, on dehydrated agar plates by Eberlin et al. [126]. In 2021, Nie et al. revealed the spatial distribution of phytochemicals in the dried root of isatis indigotica by using AP-MALDI [127].

Revealing the distribution and metabolism of drugs in organisms is crucial for the assessment of drug efficacy, safety and toxicity during drug discovery. MSI can provide the localization information of drugs and metabolites in a single experiment and has been applied to localize drug distribution, including clozapine [13], AZD2811 [128], AZD4017 and AZD8329 [129], chloroquine [130], haloperidol [131], lidocaine [132], propranolol [133] and amitriptyline [134]. In 2020, in order to study the nephrotoxicity of aristolochic acids (AAs), a rat kidney treated with AAs was analyzed by AFADESI and a significant change of 38 metabolites associated with the arginine–creatinine metabolic pathway, the serine synthesis pathway, urea cycle and the metabolism of choline, lipids, lysine, histamine and adenosine triphosphate was found (Figure 7E) [135]. In 2020, Zhang et al. evaluated the tumor-targeting efficiency of paclitaxel (PTX) and its prodrug (PTX-R) in three treatment groups, including the PTX injection group, PTX-liposome group and PTX-R treatment group using AFADESI-MSI (Figure 7F). Both PTX-R and metabolized PTX were found to be mainly accumulated in the tumor tissue with a high targeting efficiency in the PTX-R group [136].

Quantification is another important issue in AMSI. The mimetic tissue model [137] and deposition [138] are two commonly used methods for quantitative analysis in MSI. In 2016, Hansen et al. added the isotope internal standard (IS) of amitriptyline in tissue homogenates of lung, kidney, liver and heart tissues to compensate for the difference of intensities caused by different ion suppression effects and extraction efficiency. Their experiments showed the importance of preparing the standard curve in the same matrix as the unknown sample whenever possible and similar quantitative results were provided by spiked tissue homogenates and droplet deposition (Figure 7G) [139]. Song et al. introduced a new quantitative method termed virtual calibration (VC) MSI for accurately quantifying analytes across heterogeneous biological tissue. Analyte response-related endogenous metabolite ions were chosen as native internal standards to solve the difficult problem of evenly adding the IS in tissues. This method was demonstrated to be suitable for accurately mapping the quantity of analytes in the whole body and the linearity of the standard curve calibrated with VC was as good as using isotopic IS (Figure 7H) [140].

**Figure 7 metabolites-11-00780-f007:**
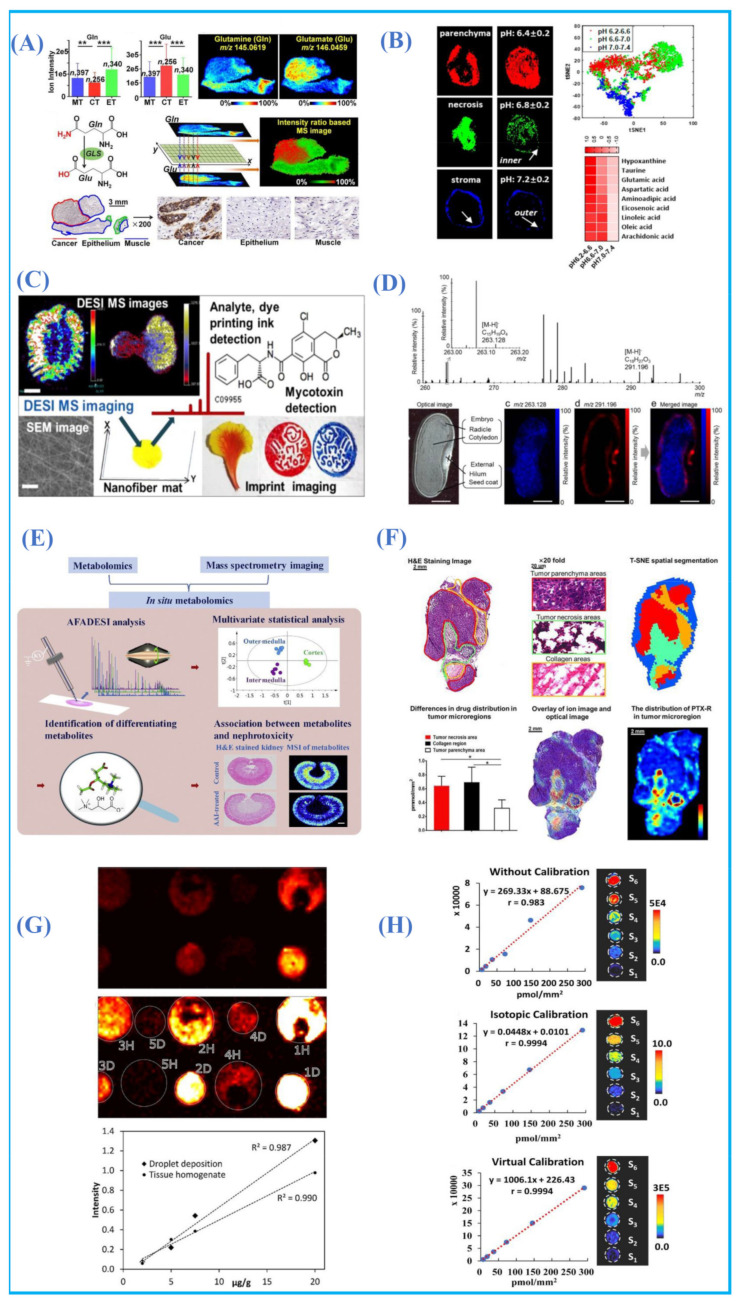
(**A**) In situ visualization of crucial metabolites and metabolic enzymes in the glutamine metabolism pathway. [117] Copyright (2019) National Academy of Sciences. (**B**) Images of the acidic TME and contributive acidic species. Reprinted with permission from the authors of [120]. Copyright (2021) American Chemical Society. (**C**) Electrospum nanofiber mats as “smart surfaces” for MSI and imprint imaging. Reprinted with permission from the authors of [122]. Copyright (2016) American Chemical Society. (**D**) MS spectrum, optical image and MSI imaging of immature phaseolus vulgaris L. seed. Reprinted with permission from the authors of [123]. Copyright (2017) Springer Nature. (**E**) In situ metabolomics in nephrotoxicity of aristolochic acids in rat kidneys. Reprinted with permission from the authors of [135]. Copyright (2020) Chinese Pharmaceutical Association and Inetitute of Materia Medica, Chinese Academy of Medical Science, production and hosting by Elsevier B.V. (**F**) Intratumoral distribution of PTX-R with heterogeneous characteristics. Reprinted with permission from the authors of [136]. Copyright (2020) the author(s) and published by Ivyspring International Publisher. (**G**) Calibration curves based on tissue homogenates and droplet deposition. Reprinted with permission from the authors of [139]. Copyright (2016) American Chemical Society. (**H**) Standard curves obtained with different calibration methods. Reprinted with permission from the authors of [140]. Copyright (2019) American Chemical Society.

## 4. Conclusions and Further Perspectives

In the past 10 years, AMSI techniques have shown unique advantages in clinical, life science, medicine, environment, forensic medicine, metabolomics and lipidomics for its characteristics of in situ, real-time, fast and high throughput. In this review, we introduce the commonly used AMSI methods and highlight the recent applications of AMSI in lipid and metabolite analysis in the last five years. (1) A comprehensive and exhaustive analysis of single cells help to provide accurate chemical information of cells and elucidate the relationship between the function of cells and chemical composition. A few micrometers resolution has been obtained in AMSI and thus a further improvement of spatial resolution and sensitivity will be vitally important for single-cell imaging. (2) Although various quantitative mass spectrometry imaging methods based on the mimetic tissue model or deposition, have been developed to evaluate the drug efficacy and pollutant toxicity, the development of universal and precise quantitative methods in AMSI is still highly expected. (3) Accurate identification and imaging of compound isomers is another challenge in ambient conditions. Recently, great efforts have been made for the AMSI of lipid isomers with different double-bond positions with the assistance of ultraviolet photodissociation or CID. In the future, due to its favorable advantages of portability and versatility, AMSI has been able to be combined with more techniques such as ion mobility mass spectrometry for isomer discrimination. (4) In recent years, emerging algorithms, such as machine learning, have made the discovery of disease markers more accurate, objective, convenient and intelligent. Thus, the development and establishment of practical and effective statistical methods, software and databases will help to extract valuable information from a large amount of raw data and will be an active research field in AMSI. (5) Owing to its ambient operation condition and fast acquisition rate, AMSI has promising application potential within in vivo mass spectrometry imaging, especially during surgery. Taken together, these improvements will keep the AMSI techniques at the cutting edge and become a more powerful and versatile tool in the analysis of complex biological samples. We believe that AMSI techniques have great application prospects in 3D imaging, protein imaging, lipidomics and metabolomics, clinical diagnosis, precision medicine, toxicology and drug metabolism in the near future.

## Figures and Tables

**Figure 1 metabolites-11-00780-f001:**
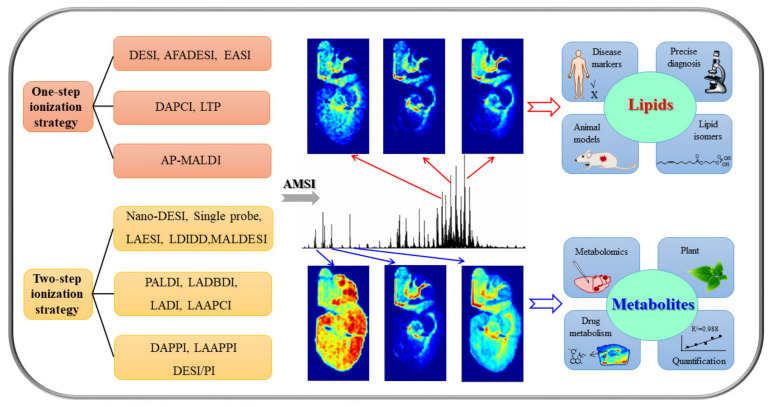
The schemes of ambient mass spectrometry imaging for lipid and metabolite analysis. DESI: desorption electrospray ionization; AFADESI: airflow-assisted desorption electrospray ion-ization; EASI: easy ambient sonic spray ionization; DAPCI: desorption atmospheric pressure chemical ionization; LTP: low-temperature plasma; AP-MALDI: atmospheric pressure matrix-assisted laser desorption/ionization; Nano-DESI: nanospray desorption electrospray ionization; LAESI: laser ablation electrospray ionization; LDIDD: laser desorption/ionization droplet delivery; MALDESI: matrix-assisted laser desorption electrospray ionization; PALDI: plasma-assisted laser ionization; LADBDI: laser ablation dielectric barrier discharge ionization; LADI: laser ablation direct analysis in real time; LAAPCI: laser ablation atmospheric pressure chemical ionization; DAPPI: desorption atmospheric pressure photoionization; LAAPPI: laser ablation atmospheric pressure photoionization; DESI/PI: desorption electrospray ionization/postphotoionization.

**Figure 4 metabolites-11-00780-f004:**
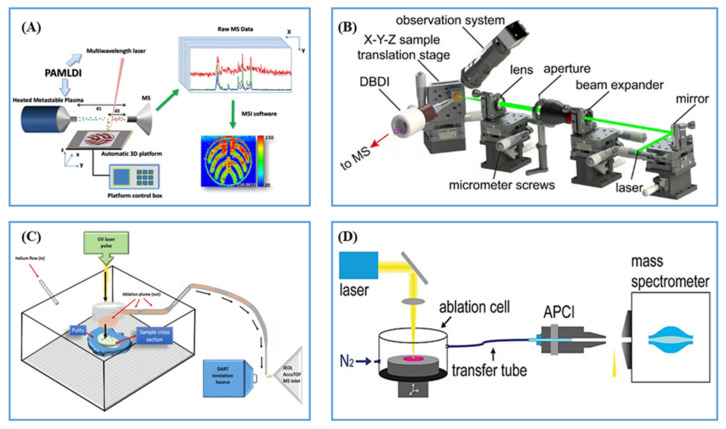
(**A**) Scheme of PALDI ion source. Reprinted with permission from the authors of [60]. Copyright (2014) American Chemical Society. (**B**) Scheme of LADBDI ion source. Reprinted with permission from the authors of [61]. Copyright (2021) American Chemical Society. (**C**) Scheme of LADI ion source. Reprinted with permission from the authors of [63]. Copyright (2017) American Chemical Society. (**D**) Scheme of LAAPCI ion source. Reprinted with permission from the authors of [64]. Copyright (2013) John Wiley and Sons Ltd.

**Figure 5 metabolites-11-00780-f005:**
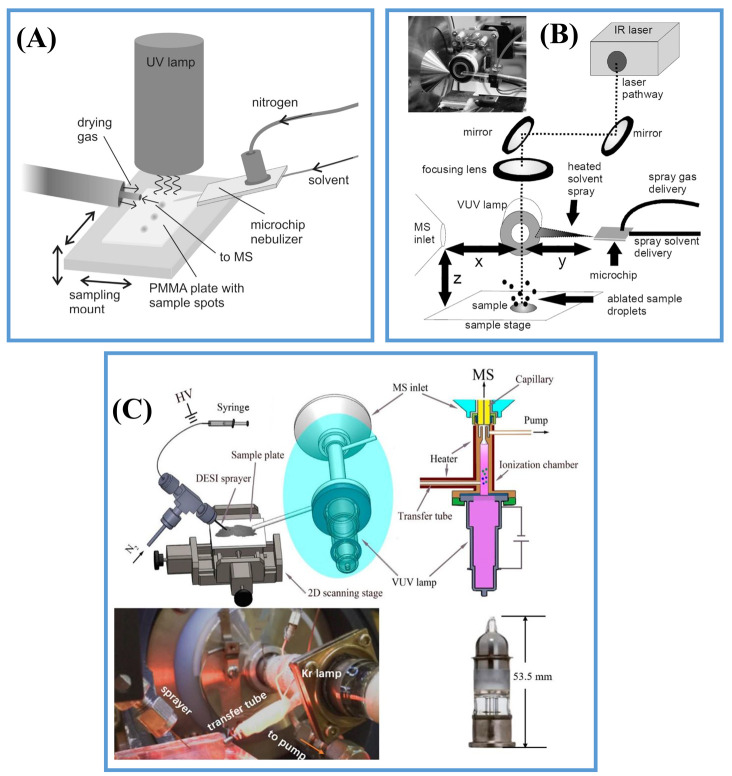
(**A**) Scheme of DAPPI ion source. Reprinted with permission from the authors of [66]. Copyright (2007) American Chemical Society. (**B**) Scheme of LAAPPI ion source. Reprinted with permission from the authors of [69]. Copyright (2012) American Chemical Society. (**C**) Scheme of DESI/PI ion source. Reprinted with permission from the authors of [73]. Copyright (2019) American Chemical Society.

**Table 1 metabolites-11-00780-t001:** Summary of one-step desorption/ionization for AMSI.

Methods *	Desorption/Ionization	Advantages	Resolution (µm)
DESI	Liquid/Electrospray	Widely used, soft ionization	50–200 (35)
AFADESI	Liquid/Electrospray	Medium sensitivity and soft ionization	100–200 (35)
EASI	Liquid/Sonic spray	No high voltage, similar to DESI	100–200 (50)
DAPCI	Plasma/Corona	High sensitivity for trace compound	200–500 (58)
LTP	Plasma/Corona	Medium resolution and sensitivity	200
AP-MALDI	Laser/Laser	High resolution and reproducibility	1.4

* The abbreviation of AMSI techniques. DESI: desorption electrospray ionization; AFADESI: airflow-assisted desorption electrospray ionization; EASI: easy ambient sonic spray ionization; DAPCI: desorption atmospheric pressure chemical ionization; LTP: low-temperature plasma; AP-MALDI: atmospheric pressure matrix-assisted laser desorption/ionization.

**Table 2 metabolites-11-00780-t002:** Summary of two-step desorption/ionization for AMSI.

Methods *	Desorption/Ionization	Advantages	Resolution (µm)
Nano-DESI	Liquid/Electrospray	High resolution and sensitivity	12–100 (12)
Single probe	Liquid/Electrospray	High resolution and sensitivity	10–20 (8.5)
LAESI	Laser/Electrospray	High resolution and matrix free	70
MALDESI	Laser/Electrospray	High resolution and sensitivity	10
LDIDD	Laser/Electrospray	High resolution and sensitivity	2.4–3
PALDI	Laser/Plasma	High resolution and sensitivity	40–60
LADBDI	Thermal/Corona	High resolution and sensitivity	20
LADI	Laser/Plasma	High resolution and sensitivity	50
LAAPCI	Laser/Corona	Medium resolution	100
DAPPI	Liquid/PI	Low resolution and soft ionization	1000 (700)
LAAPPI	Laser/PI	Low resolution and high repeatability	44–400
DESI/PI	Liquid/ESI-PI	High sensitivity and soft ionization	~200

* The abbreviation of AMSI techniques. Nano-DESI: nanospray desorption electrospray ionization; LAESI: laser ablation electrospray ionization; MALDESI: matrix-assisted laser desorption electrospray ionization; LDIDD: laser desorption/ionization droplet delivery; PALDI: plasma-assisted laser ionization; LADBDI: laser ablation dielectric barrier discharge ionization; LADI: laser ablation direct analysis in real time; LAAPCI: laser ablation atmospheric pressure chemical ionization; DAPPI: desorption atmospheric pressure photoionization; LAAPPI: laser ablation atmospheric pressure photoionization; DESI/PI: desorption electrospray ionization/postphotoionization.

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
