# Peer review of "Recent Advances of Ambient Mass Spectrometry Imaging and Its Applications in Lipid and Metabolite Analysis"

_metabolites, 2021, doi:10.3390/metabo11110780_

Round 1
Reviewer 1 Report
In the review entitled "Recent Advances of Ambient Mass Spectrometry Imaging and its Applications in Lipids and Metabolites Analysis" the authors led by Qi have done a very nice job summarizing the work in the field. The manuscript is also very well written. Although I think it's great, there are a couple things that would add to the manuscript as detailed below (numbers refer to lines):
3 Lipids and metabolites should be singular
20 hundreds or thousands is often cited for MSI but as one who does this regularly, thousands of compounds is often a stretch in a single experiment. usually, anything close to a thousand includes isotopes, non-biological features, or contaminants. I would clarify this statement.
33-35 There are a ton of acronyms used which reduce the comprehension of the sentence. There are issues with this throughout. I would try to limit the use of acronyms whenever possible and don't define acronyms if they are used a single time.
70 It's present here, but an issue elsewhere. The authors pick nondescriptive words like "big", "good", or "high". As an analytical paper, it would be helpful to have some more descriptove terms or concrete numbers. In this case, what sort of improved sensitivity would be helpful? An order of magnitude? Two or three orders of magnitude? Alternatively, they could indicate what sort of increase is required to match SIMS or MALDI. Line 106 is another example
76-79 Sentence fragment
114-115 in comparison to what exactly? Microscopy and spectroscopy both can do significantly better than this.
137 typo
166-169 The authors discuss that water reduces interferences, but do not mention diffusion at all, which is particularly problematic for metabolites.
177 MALDI-2 may be an interesting avenue for the authors to comment on.
234 See above
246 spell out 2
258 Define acronyms in table to make it more readable. This table would benefit from numbers in advantages. For example "better resolution" seems to span 20-100 µm which is innappropriate to me.
261-271 It would be worth adding that few other technologies can image lipids, unlike proteins.
298-303 It always irritates me when the main push for ambient imaging is cancer boundaries during surgery. While I understand the concept, it is amazingly difficult to get new tools approved within an operation room. It would be helpful to discuss this as well as add additional reasons why these analyses are important. If the sole reason is in an operation room, it reduces the impact of this research because of the shear difficulty getting new technology into a surgical room.
380 neurotransmitter is a very broad class of metabolites but is next to two very specific examples
380-391 Small metabolite analysis is incredibly important and difficult area of reserach. This is largely because they are fairly easy to bias with sample preparation as well as their propensity to quickly degrade. This is especially true of fatty acids. MALDI approaches circumvent this issue but having a matrix applied that is an antioxidant and a preservative. SIMS can get around this by using high vaccuum. Ambient approaches do not really account for this, from what I have seen. The authors should comment on this if they keep the section on metabolites.
418-438 The paragraph is a giant list.
493-500 The authors end the manuscript with very negative statements. While true, it really undermines the field when put at the end. It would be more helpful if solutions were added or if these were integrated earlier in the paper to leave room for perspective on the positives.
While most of this review focuses on what the authors could improve, I will say that the review is very well written and comprehensive. It is also diverse on the authors that are cited, which I appreciate. I really think the authors did a good job as an initial submission.
Excited to see the final draft.
Reviewer 2 Report
This is an important area of research and the authors attempt to capture the main contributions to the field. I have three major comments/concerns
- The authors use the figures with permission from other articles - which is common in review articles. However, they did not really make any new figures to really define what has been accomplished and how all these different approaches fit together. It was more of a compilation of techniques - albeit well organized - but did not really describe the accomplishments at any level of detail that allows the reader to see what has been accomplished.
2. The authors talk about techniques and even show figures from older techniques that have not really furthered the field (e.g., ELDI figure from 2007). This technique, in my opinion, should not be included, because nothing in the recent past shows any evidence that this is the approach to take for imaging mass spectrometry. Contrary to that, a recent review article came out on MALDESI (Mass Spectrometry Reviews, 2021, https://doi.org/10.1002/mas.21696). A quick scan of that extensive review reveals that this technique has made major advancements in instrumentation and laser technology, stage control, measurement of lipids and metabolites, including neurotransmitters, 3D imaging, etc. yet it was barely mentioned in the review. Thus, the review seems to be very biased as to what was included and is not representative of the field.
3. There is a key section missing in this review, what does the future hold? Predictions for the field (not a technique in particular) as a whole? Where will its biggest impact be made? What needs to happen technologically etc. for those impacts to be realized. An authoratative conclusion / prediction would be nice.
Round 2
Reviewer 2 Report
Revision was done to improve the review. Accept as is.